# Diminished Ovarian Reserve in Endometriosis: Insights from In Vitro, In Vivo, and Human Studies—A Systematic Review

**DOI:** 10.3390/ijms242115967

**Published:** 2023-11-04

**Authors:** Zhouyurong Tan, Xue Gong, Chi Chiu Wang, Tao Zhang, Jin Huang

**Affiliations:** 1Department of Obstetrics and Gynaecology, Faculty of Medicine, The Chinese University of Hong Kong, Hong Kong SAR, China; zyrtan@link.cuhk.edu.hk (Z.T.); gongxuexl@link.cuhk.edu.hk (X.G.); ccwang@cuhk.edu.hk (C.C.W.); 2Department of Obstetrics and Gynaecology, The Second Affiliated Hospital, The Chinese University of Hong Kong, Shenzhen 518172, China; 3School of Biomedical Sciences, Faculty of Medicine, The Chinese University of Hong Kong, Hong Kong SAR, China; 4Chinese University of Hong Kong-Sichuan University Joint Laboratory in Reproductive Medicine, The Chinese University of Hong Kong, Hong Kong SAR, China; 5Reproduction and Development, Li Ka Shing Institute of Health Sciences, The Chinese University of Hong Kong, Hong Kong SAR, China

**Keywords:** endometriosis, ovarian reserve, treatment, target therapy, ovarian preservation

## Abstract

Endometriosis, a prevalent disorder in women of reproductive age, is often associated with undesired infertility. Ovarian reserve, an essential measure of ovarian function that is crucial for maintaining fecundity, is frequently diminished in women with endometriosis. Though the causative relationship between endometriosis and reduced ovarian reserve is not fully understood due to the lack of standardized and precise measurements of ovarian reserve, there is ongoing discussion regarding the impact of interventions for endometriosis on ovarian reserve. Therefore, in this review, we investigate articles that have related keywords and which were also published in recent years. Thereafter, we provide a comprehensive summary of evidence from in vitro, in vivo, and human studies, thereby shedding light on the decreased ovarian reserve in endometriosis. This research consolidates evidence from in vitro, in vivo, and human studies on the diminished ovarian reserve associated with endometriosis, as well as enhances our understanding of whether and how endometriosis, as well as its interventions, contribute to reductions in ovarian reserve. Furthermore, we explore potential strategies to modify existing therapy options that could help prevent diminished ovarian reserve in patients with endometriosis.

## 1. Introduction

Endometriosis is a prevalent gynecological condition; it is defined by the growth of endometrial-like tissue outside the uterine cavity, and it affects an estimated 10% of women during their reproductive years [1]. This disorder is often associated with a spectrum of symptoms, including pelvic pain, irregular menstrual cycles, and—notably—infertility [2]. The complex etiology behind endometriosis-associated infertility remains an area of intense research and discussion, particularly with respect to the role of diminished ovarian reserve (OR) [2].

OR has long been defined as the number and quality of oocytes, which determine the reproductive potential a woman possesses [3]. Initially, OR was understood to encompass both the quantity and quality of a woman’s oocytes. However, the most updated conception of OR has refined this concept; it now distinctly refers to OR as the total count of the remaining oocytes. Essentially, it is a measure of oocyte quantity that is distinct from the potential of a fertilized oocyte leading to a live birth (which thus pertains to the oocyte quality) [4].

Crucial to the understanding of OR is that the vast majority of these reserves are primordial follicles [5]. These structures remain dormant until individual follicles are recruited for maturation. It is a continuous process characterized by the transition of non-growing follicles into growing follicles, which subsequently either reach maturity and ovulation or undergo atresia. Normally, a single follicle achieves full maturation to release an oocyte during each menstrual cycle, and this is regulated meticulously using hormones such as gonadotropins [6]. The importance of OR extends into various facets of reproductive medicine [7,8]. It not only informs the biological timeline for a woman’s fertility, but it also serves as a critical parameter in assessing and planning the management of infertility. Specifically, evaluating OR can guide personalized treatment strategies for infertile couples and inform procedures like fertility preservation [9].

However, it is well documented that certain pathological conditions, including endometriosis, disrupt normal follicular activation and maturation, thereby contributing to ovarian dysfunction and a consequential decline in OR. Women with endometriosis, particularly those of advanced age, are at a heightened risk of reduced OR compared to their younger, healthy counterparts [7,8]. The mechanisms instigating this decline—particularly in the context of endometriosis—are not yet fully elucidated, and they continue to be the subject of comprehensive scientific inquiry. Amidst the multifactorial dynamics leading to infertility in endometriosis, one emerging concern is whether the disease itself or the interventions employed for its management contribute to the diminishing OR. The medical and surgical treatments, though necessary for symptom alleviation and improving life quality, raise questions about their potential impact on the OR, especially given that certain procedures may inadvertently compromise OR [10].

This systematic review is designed to delve into the current understanding of OR within the context of endometriosis. The review involves drawing on recent in vitro, in vivo, and human studies that explore the mechanisms through which endometriosis and its treatments may precipitate changes in OR. Using this comprehensive analysis, we aim to unravel the impact of interventions for endometriosis on OR and suggest potential strategies to modify existing therapies to prevent diminished OR in patients with endometriosis. In detail, the aims are to (1) describe the existing prediction methods of ovarian reserve, as well as evaluate the advantages and disadvantages of each method; (2) elucidate the relevance of endometriosis per se and its treatments and reduced OR, as well as reveal the potential mechanisms; (3) highlight the potential therapeutic interventions for preserving OR in patients with endometriosis.

## 2. Methods

This review adheres to the guidelines stipulated using the PRISMA (Preferred Reporting Items for Systematic Reviews and Meta-Analyses) methodology [11]. A comprehensive literature search was conducted using the PubMed and Embase databases in July 2023. The search strategy employed (to accrue relevant publications) was as follows: (((In vitro[MeSH Terms]) OR (animal model[MeSH Terms])) OR (“Clinical Study” [Publication Type])) AND ((“Endometriosis”[Mesh]) AND (((Ovarian reserve[MeSH Terms]) OR (fertility index[MeSH Terms])) OR (Fertility[MeSH Terms]))). An additional ten pertinent articles were sourced from the references contained within other reviews. The selection of articles was undertaken independently by two authors, and any disputes were resolved under the arbitration of a senior author. The initial screening involved examining titles and abstracts to identify the studies that discussed the impact of endometriosis on fertility or ovarian reserve for in vitro, in vivo, or in clinical settings. Post the exclusion process of the reviews, non-relevant publications, duplicates, and conference papers, a total of 21 articles were earmarked for detailed analysis. A flow diagram of the study selection process is shown in Figure 1.

## 3. Results and Discussion

### 3.1. Characteristics of Studies on the Effect of Endometriosis and Its Therapeutic Interventions on Ovarian Reserve

The key characteristics of the studies conducted in in vitro, in vivo models, and humans are summarized in Table 1. Out of the 21 studies included, 10 were conducted on humans, 11 were conducted on in vivo models, and among these studies, 4 continued their investigations using in vitro models. Peritoneal endometriosis is the most frequently studied subtype in in vivo models due to its standardized methodology, while endometrioma (OMA) is the most common subtype in clinics [12]. As endometriosis can have varying effects on OR and its underlying mechanisms depending on the subtype, it raises awareness that different subtypes of in vivo models should be applied to explore this topic thoroughly. In in vivo models, the most common measurement of OR is follicle count, and in human studies, anti-Müllerian Hormone (AMH) levels and antral follicle count (AFC) were extensively utilized for OR assessment. The further analysis based on those studies is thoroughly described in the following contents. 

### 3.2. The Prediction of Ovarian Reserve

Although the definition of OR was updated in 2020, the existing prediction methods of OR are still based on the traditional conception. The assessment of OR can be categorized into quantitative and qualitative parameters. The quantitative markers indicate the number of follicles and oocytes, whereas qualitative markers represent the oocyte quality during folliculogenesis and post-ovulation.

#### 3.2.1. Quantitative Assessment

In clinical practice, the prediction of OR can be achieved using AFC using high-quality transvaginal ultrasound [34], which is non-invasive, inexpensive, and easy to operate. However, it should be measured at a specific time in the menstrual cycle, and the widespread utilization and uniform application of this method is hindered by its dependence on the operator and equipment, leading to inconsistencies across different regions and institutions [35]. Specifically, in women with unilateral OMA, AFC could better reflect OR of the affected ovary. Whereas the AFC may be underestimated in ovaries with an OMA due to the stretch of the ovary and distortion of pelvic anatomy [32,36], which increases the distance from the probe to the ovary and impairs the resolution of transvaginal ultrasound [36,37]. Alternatively, the serum level of follicle-stimulating hormone (FSH) serves as a conventional endocrine marker for OR and is more cost-effective [38]. FSH is synthesized and secreted by the anterior pituitary gland. This hormone acts upon granulosa cells (GCs), promoting them to metabolize androgens, secreted by theca cells, into estrogen within maturing ovarian follicles. Concurrently, as the follicles grow, there’s an observed escalation in estrogen produced by the GCs [39]. Elevated serum levels of FSH have been correlated with a diminished reserve of primordial follicles and a heightened proportion of actively developing follicles. Given the pivotal role of the primordial follicle pool in defining OR, increased FSH concentrations in serum are specific but not sensitive indicators of a compromised OR in a clinical context [40]. Additionally, FSH, in conjunction with luteinizing hormone (LH), functions as a gonadotropin instrumental in modulating the menstrual cycle [41]. Its concentrations exhibit cyclic fluctuations: peaking during the menstrual and ovulatory phases and decreasing during the late follicular and luteal phases [42]. Consequently, when leveraging serum FSH as a prognostic marker for the prediction of OR, it requires special timing to carry out the test, usually on days 2 to 4 of the menstrual cycle. However, its utility as a consistent marker for OR is compromised due to the inter-cycles’ variability [43,44]. Another marker of OR is serum AMH, which is secreted from GCs in pre-antral and antral follicles up to 8 mm in diameter [45], and follicles of 4–8 mm in diameter contribute about 60% of the circulation AMH [46]. AMH has a role in regulating ovarian function during the early stage of follicle development by inhibiting the excessive recruitment of primordial follicles and delaying the transition of these follicles to the primary stage, thus ensuring a maintained follicle pool [47]. Unlike FSH, the level of AMH remains stable throughout the menstrual cycle [48,49]. Moreover, Weenen’s team has demonstrated the consistency of AMH to AFC [34,45,50], while AMH is more sensitive than FSH and tends to decline before FSH rises [51]. However, the widespread utilization of AMH is limited due to technical challenges such as standardizing commercially available kits and ensuring sample stability under variable store conditions [52,53]. FSH and AMH are indicators of OR of both ovaries, not just one.

The most direct assessment method of OR is follicle count using histological analysis [54,55]. In this procedure, ovarian tissue is embedded in paraffin, sectioned, and stained with hematoxylin and eosin or Periodic acid Schiff hematoxylin. Follicles are classified into different stages based on their morphology, and the total number of follicles across all sections is calculated as the OR [56]. Although this method provides an accurate measure of follicles that directly reflects OR, it is a labor-intensive process that requires the use of the entire ovarian tissue and has limited potential for clinical application. A high-throughput ovarian follicle-counting method utilizing deep learning approaches has been proposed for detecting and counting follicles in mouse ovaries. However, it still involves tissue sampling, which is time-consuming [57]. Moreover, follicle distribution can be calculated based on non-invasive sonography, which could provide additional evidence of follicle maturation [15].

Furthermore, in both clinical and research settings, the number of mature oocytes collected following superovulation can serve as an indicator of OR [58,59]. However, the limitations of this approach include its dependence on the stimulation protocol and the intracycle variability of ovarian response [44].

Overall, the current quantitative measurements for OR are limited by technical factors such as fluctuations in FSH levels and operational bias of AMH and AFC. Either AFC or AMH reflects only a portion of the overall OR landscape. In animal models, follicle count can provide more accurate measurement using histological analysis, but this comes at the cost of laborious procedures and the consumption of the entire ovary. The quantitative assessment of ovarian reserve is summarized in Figure 2. Additionally, the comparison of the existing qualitative assessments in clinics is listed in Table 2.

#### 3.2.2. Qualitative Assessment

The limited pool theory postulates that quantitative reduction in OR may lead to a decline in the quality, characterized by impaired oocyte development and decreased oocyte quality [44]. The natural fertility of women and the successful rate of assisted reproductive technologies (ART) can reflect OR, including time to achieve pregnancy, fecundity, miscarriage rate, and genetic defects post-delivery [43,60]; whilst, in ART, in-vitro fertilization (IVF) rate, blastocyst formation, embryo morphology, and implantation failure rate can reflect OR qualitatively. Oocyte and embryo quality are considered more likely to be indicated by pregnancy rate and live birth rate [61,62].

From a research perspective, sixteen genes, including GRM3, UTS2, FAM9B, and LYG2, which encode secretory proteins, have been identified for their unique expression in the oocytes of primordial and primary follicles. Their specificity in these early developmental stages suggests a pivotal role in the foundational processes of follicular development, which is a crucial determinant of the quality of OR. Moreover, the nature of the proteins they encode potentially alludes to their involvement in intricate cellular signaling mechanisms or other vital processes that underpin the health, viability, and functionality of the oocytes. This is in stark contrast to merely quantifying follicles. Their prospective utility of biomarkers in predicting OR in cell culture systems further underscores their potential significance in assessing follicular quality rather than sheer quantity. However, further validation via rigorous research is required [63]. Additionally, the metabolic characteristics and cytokines profiles in the follicular fluid (FF) and peritoneal fluid (PF) might impact folliculogenesis and OR, holding potential as future markers for assessing OR [64,65]. The qualitative assessment of ovarian reserve is summarized in Figure 3.

### 3.3. The Link of Endometriosis with Reduced Ovarian Reserve

#### 3.3.1. The Clinical Relevance of Endometriosis and Ovarian Reserve from Human Studies

##### Endometriosis Per Se

Endometriosis is associated with reduced OR, especially in patients with OMA, where the endometriotic lesions implant in the ovarian tissue [66]. It was reported that affected ovaries with OMA had lower AFC compared to the contralateral healthy ovaries [66,67]. Moreover, reduced serum AMH levels have been reported in women with OMA compared with those without OMA [68]. The patients with bilateral OMA had lower AMH levels than the unilateral ones, and the OMA size is negatively correlated with AMH levels, suggesting that OMA itself has a detrimental effect on OR [27]. These findings are further confirmed using a prospective longitudinal study [29], in which the authors found that the AMH levels in women with OMA decrease faster than those in healthy women. During the 6-month follow-up period, the AMH levels declined 26% in the OMA group, compared to 7% in the control group, and the rate of decline in AMH levels was relevant to age.

##### Endometriosis Treatments

Endometriosis is a chronic inflammatory disease requiring lifelong management [69]. Treatment of endometriosis is aimed at suppressing lesion growth, relieving pain, promoting fertility, and treating the systematic effects of the disease [70,71]. Current therapeutic options for endometriosis include medical and surgical treatment. In recent years, there has been growing concern about how endometriosis treatment affects women’s OR.

Medical Treatment

Non-Steroidal Anti-Inflammatory Drugs (NSAIDs)

NSAIDs are a group of medicines applied to alleviate pain, reduce inflammation, and lower fever. They achieve this by blocking the action of cyclooxygenase (COX) enzymes, specifically COX-1 and COX-2. NSAIDs are frequently employed to relieve endometriosis-associated pain, either alone or in combination with other treatments [72]. However, the available reports on the effects of NSAIDs on OR in endometriosis are very scarce. It was reported that the use of NSAIDs might prevent ovulation in women [73,74], and selective COX-2 inhibitors and NSAIDs with a long half-life tend to inhibit ovulation more intensely [75]. The COX-2-dependent prostaglandins seem to mediate ovarian collagen degradation by promoting the generation of proteolytic enzymes that rupture the follicles [76]. Moreover, non-selective NSAIDs with a brief half-life may allow adequate COX production at drug-free intervals. In a double-blind crossover study, 22 healthy women were enrolled and orally administered meloxicam (a selective COX-2 inhibitor) for five consecutive days in the late follicular phase. This treatment commenced when the leading follicle reached a diameter of 18 mm, with a dosage of either 15 or 30 mg/day [75]. The results showed a higher incidence of dysfunctional ovulation in cycles treated with 30 mg/day (90.9%) than those treated with 15 mg/day (50%), indicating dose dependency of ovulation inhibition. Additionally, using NSAIDs for the first eight days of the menstrual cycle does not inhibit ovulation compared to using them continuously until midcycle [75,77]. However, the underlying mechanisms of prostaglandins regulating ovulation are only partially understood. So far, there is insufficient evidence to conclude the effect of NSAIDs on OR in women with endometriosis. Hence, further research, especially randomized controlled trials with well-stratified patient groups, is warranted to generate compelling evidence. For patients who desired pregnancy recently, the administration of NSAIDs should be carefully considered, as they may have a potentially detrimental effect on conception.

Hormone Treatments

Combined oral contraceptives (COC) and progestogens represent the first-line therapy to reduce endometriosis-related pain [78,79]. COC combines an estrogen component with a different generation of progestin components, producing a negative feedback effect on the hypothalamic-pituitary-ovarian (HPO) axis [80], suppressing ovarian activity, and reducing estrogen-induced production of prostaglandins and inflammation [81]. Progestin eases endometriosis-related symptoms by inducing endometrium decidualization and inhibiting ovulation, leading to thinner endometrium and lesion atrophy [82]. Women who had used COC for more than a year had significantly less AFC as well as a smaller ovarian volume compared with the control group [83]. Regarding the effect of hormonal contraceptives on AMH, there are still controversies. The impact of current hormonal contraceptive use on AMH has been examined using longitudinal [41,84,85,86] and cross-sectional [87,88,89] studies, with the decrease in AMH levels varying from no different to 55% in contraceptive users. A cross-sectional cohort study including 27,125 women compared serum levels of AMH in contraceptive and non-contraceptive users [89]. The results showed that AMH levels were significantly lower in women using COC, vaginal ring, hormonal intrauterine device, implant, or progestin-only pill than in non-contraceptive users. Additionally, the relationship between contraceptive use duration and AMH level remains unclear. However, another small randomized controlled trial (RCT) showed that AMH levels are unaffected by one month of exogenous sex steroids used for contraception, whether administered orally or vaginally [41]. The different results may be interpreted by the small sample size, individual variations, the duration of hormonal contraception, and the various time points of testing in different studies [31,90]. The decrease in AFC and AMH could be explained by hormonal contraception suppressing the HPO axis, resulting in a decreased secretion of FSH and LH, followed by inhibition of estradiol production, follicular growth, and ovulation [80,91,92]. The adverse effects of hormonal contraception on OR parameters are temporary and tend to be restored 2–3 months after discontinuation [85,93], which may be explained by the hypothesis that normal folliculogenesis takes approximately three months [94]. Although there is no evidence that COC and progestin have long-term detrimental effects on OR and fecundity, it is essential to be aware of the impact of hormonal contraception on OR tests when interpreting the results among those women. In COC users, OR tests are recommended to be performed at least three months after the discontinuation of COC.

Dienogest, a fourth-generation progestin, is widely employed for treating endometriosis-related pain [95,96]. A previous study showed that dienogest could improve IVF outcomes and reduce lesion size in patients with OMA [26], which may be attributed to its anti-inflammatory and anti-angiogenesis activity [96]. Among patients with OMA ≥ 4 cm, AFC, the number of retrieved oocytes, two-pronuclear embryos, and blastocysts were significantly higher in the dienogest treated group compared with the non-treated group, and there was no significant difference observed in terms of AMH levels. Similarly, Muller and colleagues found that six months of dienogest treatment after the excision of OMA significantly improved the number of retrieved oocytes, clinical pregnancy rates, and live birth rates compared to those without hormonal treatment [30]. An RCT involving 49 women with OMA found that those who underwent a 4-month course of dienogest or gonadotropin-releasing hormone (GnRH) agonist after laparoscopic cystectomy had varying levels of AMH retention one year later [24]. Over 60% of patients in the dienogest group retained more than 70% of their AMH levels before treatment, while none in the GnRH agonist group retained their AMH levels (*p* < 0.01). A potential reason behind this is dienogest can reduce the inflammatory response during the perioperative period by downregulating inflammatory factors, such as interleukin (IL)-6 and IL-8, via progesterone receptors [97]. On the other hand, in comparison with GnRH agonists, dienogest treatment preserves the basal FSH secretion, which supports the development of primary and small antral follicles, subsequently leading to an elevation in AMH levels [98]. For women with endometriosis, dienogest may be a safe choice for managing disease and maintaining the OR. Nevertheless, additional research is required to verify these results. GnRH analogs relieve endometriosis-related symptoms by either deregulating the pituitary GnRH receptor or directly blocking the pituitary GnRH receptor, suppressing the secretion of LH, FSH, and gonadal sex steroids, leading to shrinkage of the endometrium [99]. Regarding the side-effect profile (i.e., the reduction in bone mineral density and hypoestrogenic symptoms), GnRH analogs are prescribed as second-line therapy [72]. Previous studies showed that long-term treatment with GnRH agonists did not reduce the OR regarding AMH levels [100,101]. Recently, in a clinical trial involving 200 infertile women, a placebo-controlled study randomly allocated participants to either the study group, receiving GnRH agonist treatment, or the control group, receiving a placebo, for three months before undergoing IVF [102]. The study revealed that GnRH agonist treatment results in diminished oocyte quality, even though there were no significant differences in terms of embryo quality, clinical pregnancy rates, and cumulative live birth rates. Different approaches to ovarian stimulation were compared before IVF procedures in women with stage III or IV endometriosis in another study. Specifically, the study investigated cycles stimulated with a GnRH agonist for 7 to 10 days in the mid-luteal phase (group 1), as well as those with a two-month (group 2) or three-month (group 3) interval prior to IVF. The results showed that the number of qualified oocytes and embryos was significantly lower in group 3 than in groups 1 or 2 [103]. In 2019, a meta-analysis investigated the merit of long-term GnRH agonist therapy compared to no treatment prior to standard in-vitro fertilization and intracytoplasmic sperm injection (IVF/ICSI) in women with endometriosis [104]. In light of the paucity and very low quality of existing data, the authors failed to conclude whether long-term GnRH agonist therapy impacts IVF outcomes compared to standard IVF/ICSI. GnRH antagonist has been introduced to endometriosis treatment recently, and its efficacy on pain relief has been demonstrated by 3 RCTs [105,106,107]. However, none of them reported OR markers before and after treatment. The effect of GnRH analogs on OR and oocyte quality remains uncertain. Therefore, further high-quality trials are required to definitively determine the impact and mechanism.

2.Surgery

Surgical intervention aimed at removing endometriotic lesions and dividing adhesions has traditionally played a crucial role in the treatment of endometriosis. Surgery is recommended for women who are non-responsive to medical treatment or infertility [72]. Additionally, surgery may be necessary when ovarian malignancy is suspected. Various surgical techniques are available for treating OMA, such as cystectomy, laser ablation, electrocoagulation, and drainage. European Society of Human Reproduction and Embryology (ESHER) guidelines recommend cystectomy instead of drainage and coagulation because of the lower recurrence rate and more favorable reproductive outcomes [72,108]. However, a significant reduction in OR after surgical excision of endometriotic cysts in patients with OMA has been reported in previous studies [109,110]. The stripping technique may cause the inevitable removal of the surrounding healthy ovarian tissue, which has been histologically confirmed postoperatively [111]. Either the preoperative medical treatment or the surgical technique of surgeons might be related to the removal of normal ovarian tissue [112,113]. A multicenter RCT included 60 patients with OMA (>3 cm) who received laparoscopic cystectomy or CO_2_ laser vaporization [28]. After three months of follow-up, it was observed that the AFC of the operated ovary significantly increased in the CO_2_ laser vaporization group compared to the cystectomy group. Additionally, serum AMH levels were significantly reduced in the CO_2_ laser vaporization group, while there was no such reduction in the cystectomy group. However, the pregnancy and recurrence rates were similar after a 1-year follow-up for cystectomy and CO_2_ laser vaporization [28,114]. In addition to the surgical approach, intraoperative hemostasis also contributes to the decline in OR. It was reported that the reduction in AFC and AMH after cystectomy may be related to injury of the surrounding blood vessels by electrocoagulation [33]. A meta-analysis reported a significantly lower serum level of AMH in the coagulation group than in the suture group 6 and 12 months after surgical stripping of OMA, and the AFC in the coagulation group was significantly less than in the suture group 3 months postoperatively [115]. Another meta-analysis compared the effect of different hemostatic methods on OR following laparoscopic OMA excision; the results showed that the mean decline in serum AMH levels was 6.95% less with suturing or application of a hemostatic sealant for hemostasis than with bipolar coagulation (*p* = 0.02) at three months after surgery [116]. Moreover, the perioperative inflammatory reaction may also be involved in OR damage during laparoscopic cystectomy. A retrospective case series study reported that elevated IL-6 levels after surgery and bilateral cyst burden are the two risk factors for AMH not returning to baseline level within one year after surgery [117].

Overall, no matter what surgical techniques are performed, surgeons should be aware of the surgical-related damage to the ovary and avoid using bipolar coagulation in the residual ovaries as much as possible. Further studies are needed to evaluate which surgical technique is best for preserving OR.

#### 3.3.2. The Relevance of Endometriosis and Ovarian Reserve from In Vitro and In Vivo Studies

Currently, the main research to detect OR in patients with endometriosis is based on limited clinical markers. As mentioned above, the markers, including AFC, serum level of FSH, and AMH, can only partially reflect the OR and more likely contribute to observational study instead of studying the underlying mechanism of reduced OR and its subsequent infertility. In contrast, it is challenging to detect the accurate OR in clinics due to the obstacle of reaching the whole ovarian tissue of patients. Instead of the urgency to develop novel markers that can detect the OR with higher sensitivity and elevated stability, it’s also important to utilize experimental models for further investigations. It has been around 40 years since the development of in vitro and in vivo experimental models that recap the main characteristics of endometriosis [118]. There are multiple applications to the existing endometriosis models, including but not limited to the study of endometriosis-related infertility, in which the onset and consequence of decreased OR have been pointed out. Most of these models are used for exploring the effect of endometriosis per se on OR instead of its medical intervention. Meanwhile, the taxonomic differences should be taken into account in the related studies. More investigations based on animal models and their clinical implications should be encouraged.

##### Decreased Ovarian Reserve in In Vitro Studies of Endometriosis

In vitro models are cheap, quick, reproducible, and available to check the expression level of regulators and modify the genetic information by transfection [119]. In vitro culture systems of endometriosis can be divided into human-derived cell lines and primary cells. Both are useful in exploring hormone stasis, inflammation, proliferation, angiogenesis, and immune response involved in endometriosis pathogenesis [119]. Compared to cell lines, primary cells can better characterize the disorder. Nevertheless, their short life span impeded the research utilization [120]. Cell lines can display a longer life span and stable cellular morphology and physiology of endometriosis. The application of both primary cells and cell lines in the pathogenesis, tumorigenic transformation, and therapeutic relevance of endometriosis was summed up by Gu et al. [121]. Nonetheless, the application of in vitro endometriosis models in its associated infertility is restricted. However, the culturing system of GCs, follicles, and embryos isolated from ART should not be ignored. They could better elucidate the etiopathogenesis and novel therapeutic effects of related infertility in endometriosis [122]. It has been found that the local inflammation and extensive oxidative stress in GCs could impair follicular growth and oocyte quality, which furthermore decreases the OR [122,123,124]. Recent studies also showed the embryotoxicity of the fluids from patients with endometriosis using the embryo culturing system. When two-cell embryos were cultured with PF from patients with endometriosis, the degenerate rate of embryos was dramatically increased [125]. In another result, a significantly lower hatching rate of blastocysts was reported when the embryos were exposed to pooled sera from patients with endometriosis [126]. When bovine embryos were exposed to FF from infertile patients with mild endometriosis, a lower hate rate and defects in early embryogenesis were identified [127]. Since the hyperactivation of primordial follicles has been extensively discussed in human studies and animal models, the follicle isolation of culturing could present a direct outlook to verify the current hypothesis and investigate the mechanisms. However, due to the difficulty of follicle isolation, especially for primordial follicles, there is still a lack of related studies to explore the activation of primordial follicles and the subsequent follicular development in the presence of fluids from patients with endometriosis.

##### Decreased Ovarian Reserve in In Vivo Studies of Endometriosis

Endometriosis models range from non-human primates (NHPs) to rodents [12]. NHPs such as cynomolgus monkeys and baboons are widely applied as experimental animal models for endometriosis since they process cyclic menstruation, which enables the development of spontaneous endometriosis [128]. Similar endometriosis symptoms were found in cynomolgus monkeys in 1993 and 2018, which showed clinical relevance to those in women [129,130]. However, the prevalence of endometriosis is much lower in NHPs, and the diagnosis cannot be made until the severe development of related signs and symptoms [131,132]. The researchers, thereby, tried to induce peritoneal endometriosis artificially by using seeding and inoculation methods. The induced model showed similar phenotypes as spontaneous endometriosis while with a higher productivity [133,134]. Although NHP models manifest as the closest animal model to human beings, regarding the same reproductive system, reproductive anatomy, and immunological response, there remain defects like high expense, ethic issues, and high facility requirements, which impede their wide applications [21,135]. On the contrary, rodents are more cost-effective, easily handled, fertile, and have the availability of genetic modification, which enables their wide application in endometriosis studies [135,136,137,138]. Since rodents do not process menstruation, which hampers them to develop spontaneous endometriotic lesions, endometriosis can only be induced artificially in rodent models. The common approaches for endometrium implantation include intraperitoneal injection and suturing around the mesentery artery. Whichever method is adopted, rodents commonly manifest the subtypes of superficial peritoneal endometriosis (SUP) or deep infiltrating endometriosis (DIE) [20,139,140,141].

With the application of in vivo models of endometriosis, a decreased OR, fertility, together with poor pregnancy outcomes have been extensively reported [20,23,140,141,142]. From the research performed by Koga’s team, there is a dramatic decrease in OR in mouse models with endometriosis in comparison with sham controls. The proportion of primordial follicles was significantly diminished. Meanwhile, the growing follicles and late-staged follicles were increased, which advised a hyperactivation of primordial follicles as the potential mechanism of diminished OR [16]. Similar results have been reported by Bengochea et al. in 2016 [143]. In another result based on the Sprague Dawley mice model, the effect of endometriosis on ovarian function was affected. The researchers found out that the number of luteinized unruptured follicles was increased in mice with endometriosis, which further suggested an abnormal hormone regulation and ovulation underlying the mechanism of reduced OR in endometriosis [13]. The application of animal models merits in-depth investigations of the physiopathology of endometriosis and its related infertility with decreased OR. While the current studies in this field cannot explain the causative effect of diminished ovarian function and reduced OR.

### 3.4. Potential Mechanisms Underlying Diminished Ovarian Reserve in Endometriosis Based on In Vitro, In Vivo, and Human Studies

#### 3.4.1. Inflammation, Oxidative Stress, Fibrosis, and DNA Damage

##### In Human Studies

A significantly increased volume of PF and elevated macrophage count were found in women with endometriosis compared with the controls [144]. Levels of pro-inflammatory cytokines such as IL-1, IL-6, IL-8, and tumor necrosis factor (TNF)-α have been shown to be increased both in PF and FF in women with endometriosis [145,146,147]. The disruption of the intrafollicular environment and its effect on folliculogenesis and oocyte quality has been extensively investigated. It was reported that the concentration of TNF-α, IL-8, and IL-12 in FF is negatively correlated with oocyte quality [64,148]. As mentioned before, the perioperative inflammatory reactions could increase the level of inflammatory cytokines, which further decreases OR among women with endometriosis who have undergone laparoscopic cystectomy [117]. Additionally, from prior literature, it is mentioned that inflammation may induce excessive oxidative stress, which thereby leads to fibrosis of the ovarian cortex and further results in a reduction in OR and the DNA damage of oocytes during and after folliculogenesis [12,15,149]. Oxidative stress caused by excess reactive oxygen species (ROS) and the imbalance of antioxidants in the follicular microenvironment was regarded as important for inducing follicle senescence. It led to impaired reproductive endocrinology, disturbed follicle quantity and quality, and ended in diminished OR [150]. In addition, adhesions—whether de novo or re-formed—can potentially hinder the blood supply to the ovaries and impact the development of follicles [151,152]. The blood supply of ovaries comes from the ovarian ligament, suspensory ligament, and mesosalpinx. A retrospective cohort study revealed that either the involvement of mesosalpinx or adnexal adhesion before cystectomy in patients with OMA significantly lower serum AMH levels at one month and one year after surgery compared with those without mesosalpinx involvement, which further demonstrates that surgery may decrease the OR by disturbing in ovarian blood supply [152].

Furthermore, endometriosis patients were observed to have an accumulation of tissue inhibitors of metalloproteinase (TIMP)-1. This disrupts the usual enzyme milieu of MMP/TIMP in the peritoneal cavity, leading to adverse effects on ovarian dynamics and oocyte quality [20].

##### In Vitro and In Vivo Studies

Similar to a clinical study, when mouse embryos were cultured with PF from patients with endometriosis, an increased concentration of TNF-α was detected, which was associated with the impaired development and increased apoptosis of embryos. In the meantime, a TNF-a inhibitor was shown to alleviate the embryotoxicity [125]. These findings suggest that cytokines like TNF-α act as a potential inducer of reduced OR in endometriosis. Inhibitors targeted at these cytokines could bring the potential to reverse OR. As verified in human studies, inflammation may result in excessive oxidative stress [150]. The impact of oxidative stress on endometriosis on OR has been proved in the murine model of ovarian endometriosis [15]. The fibrosis and iron deposition were identified in endometriotic lesions, and the destroyed structure was observed in the ovarian cortex as a result of extensive ROS [15]. The vasculature is destroyed and cuts down the blood supply to follicles, eventually leading to impaired folliculogenesis and decreased follicle count [153]. In another animal study, sham rats treated with TIMP1 exhibited a decrease in the number of zygotes, ovarian follicles, and corpora lutea (CLs), as well as low-quality embryo development. These findings closely resemble the results observed in endometriosis models [19]. The finding indicated that TIMP1 is with embryotoxicity and disturbs the folliculogenesis and further leads to the diminished OR.

#### 3.4.2. Dysregulated Ovulation and Ovarian Steroids Secretion

##### In Human Studies

Ovarian function encompasses the production of high-quality oocytes for fertilization and the secretion of ovarian hormones to maintain physiological balance [154]. Numerous studies have extensively reported a significant decline in ovarian function among patients with endometriosis. It is suggested that the hyperactivation of the primordial follicle and subsequent impaired development may contribute to diminished OR and overall ovarian function, which will be further discussed in Section 3.4.3. However, another study did not observe significant differences in the number of preovulatory follicles between groups but noted a decrease in the number of ovulated oocytes in endometriosis patients [143]. Currently, due to limited published research, it remains challenging to determine whether there is a defect in ovulation and how endometriotic lesions themselves impact the ovulation process. Instead of endometriosis per se, the hormone treatment for patients with endometriosis aims to downregulate estrogen and progesterone levels, thereby inhibiting ovulation [99]. Given that ovarian steroids play a crucial role in promoting follicular growth, their dysregulated secretion is associated with diminished OR [155,156]. Altered hormone profiles have been extensively reported in individuals with endometriosis [157,158]. Although the influence of ovarian hormones on the pathogenesis of endometriosis has been investigated, the direct impact of endometriosis on ovarian steroid hormone production is unclear [159]. Women with endometriosis exhibit reduced levels of ovarian steroid hormones, including estrogen, progesterone, and androgen, in their FF [160].

##### In Vitro and In Vivo Studies

Back in 1991, it was reported that endometriosis could potentially affect ovarian function by increasing the occurrence of luteinized unruptured follicles in an NHP model of endometriosis [23]. Similarly, in a rabbit model of ovarian endometriosis, a significant decrease in ovulation point was observed in ovaries with endometrial implants [161]. Impaired ovulation is scarcely mentioned in the following years, mainly due to the ovariectomy during model establishment in mice impeding the observation of ovarian morphology and histology [162]. However, dysregulated hemostasis could reflect defective follicle development and ovulation in regard to the participation of ovarian steroids in maintaining folliculogenesis [163].

The steroid hormone estradiol assumes a vital role in the regulation of female fertility, specifically in folliculogenesis. Estradiol is locally synthesized by granulosa cells (GC) within ovarian follicles and serves as a key determinant in orchestrating the development and selection of dominant preovulatory follicles [164]. Furthermore, the significance of Estradiol and its receptors in the establishment of endometriosis has been substantiated via experimentation in a mouse model. In this model, Estradiol acts as a mediator of signaling responses via estrogen receptors, collectively exerting a pivotal influence on both the progression of endometriotic lesions and the microenvironment in which they develop [165]. In cell culture systems using endometriotic cell lines and primary cells, similar hormone imbalances have been observed [166]. The impaired balance of ovarian hormones may be attributed to defects in the hypothalamus-pituitary-ovarian axis in patients with endometriosis [167]. However, further research is needed to establish causative effects in this area.

#### 3.4.3. Hyperactivation of Primordial Follicle

##### In Human Studies

The review published by our team in 2022 concluded that endometriosis can remarkably impair the development of follicles and deteriorate the quality of oocytes by activating primordial follicles prematurely [12,168]. The role of the PI3K-PTEN-Akt-Foxo3 signaling pathway has been extensively studied over the years, revealing that PTEN inhibition acts as a major negative regulator of PI3K/AKT activity [16,169]. Loss of PTEN is associated with excessive activation of primordial follicles and subsequent follicular atresia [170]. By detecting ovarian samples from patients with endometriosis, the PI3K-PTEN-Akt-Foxo3 signaling pathway was activated [16]. It is also validated that in patients with disorders, dysregulated expression of the PI3K/AKT pathway can lead to a decreased number of follicles and oocytes, hindering follicle development, and ultimately result in reduced OR and infertility [171,172].

##### In Vitro and In Vivo Studies

An altered PI3K-PTEN-Akt-Foxo3 signaling pathway was found to excessively activate primordial follicles in a mouse model of endometriosis [16]. Recent studies suggest that mTOR activity downstream of the PI3K/AKT pathways in granulosa cells plays a central role in primordial follicle activation [172,173]. FOXO3a, a transcription factor responsible for cell cycle arrest and apoptosis, is a substrate of AKT. Interestingly, overexpression of constitutively active FOXO3 in the nucleus of mouse oocytes maintains their dormancy, while phosphorylated FOXO3a, induced by AKT signaling, prevents dormancy by being excluded from the oocyte nucleus [174]. However, it remains unclear how FOXO3a exerts these actions within the dormant primordial follicle. Andrade et al. identified a molecular landscape characterized by low PTEN, high PI3K-AKT activity, and low FOXO3a, which are associated with increased follicle activation and oocyte maturation potential [175]. The hyperactivated PI3K-PTEN-Akt-Foxo3 pathway and diminished primordial follicles were observed in a mouse model of endometriosis. In comparison, the proportion of primordial follicles was restored with the inhibitor of the PI3K-PTEN-Akt-Foxo3 pathway [16]. These indicate that the well-programmed PI3K/AKT pathways in follicles contribute to normal oocyte growth during folliculogenesis.

### 3.5. Potential Therapeutic Interventions

#### 3.5.1. Hyperactivation of Primordial Follicle

As mentioned above, either perioperative inflammatory reactions or adhesion have detrimental effects on OR in women with endometriosis. To minimize the impact on OR, measurements to reduce perioperative inflammation and prevent adhesion formation should be considered. Certain substances, such as ferric hyaluronic acid, dextran sulfate, and fibrin, effectively reduce postoperative adhesions but do not prevent their formation [176]. Research has shown that Pirfenidone, an anti-fibrotic medication, can effectively decrease postoperative adhesions in women who suffer from endometriosis [177]. A novel adhesion barrier powder made from purified potato starch has been found to effectively reduce adhesion severity and extent by 85% compared to saline solution irrigation after endometriosis surgery, as confirmed by a second-look laparoscopy [178]. After 12 months of follow-up, the pregnancy rate in the intervention group was significantly higher than that in the control group, and the cycle-independent pelvic pain and dysmenorrhea were improved in the intervention group [179]. One limitation of this study is that the researchers did not evaluate the changes in OR markers before and after treatment.

As described in Section 3.3.1, dienogest showed anti-inflammatory effects by down-regulating inflammatory factors via progesterone receptors [97]. The administration of a 4-month course of dienogest postoperatively was found to ameliorate the decrease in AMH in women who underwent laparoscopic cystectomy compared with those treated with GnRH agonist after surgery [24], indicating that dienogest may be an alternative therapeutic intervention that could reduce the lesion size of OMA and might have a lesser impact on the ovarian reserve.

#### 3.5.2. The Choice of Different Surgical Options Regarding Endometriosis

In decades, the impact of endometriosis surgery on OR has raised concerns among gynecologists regarding the use of different surgical procedures in this field. The effect of different surgical procedures on OR has been discussed above; CO_2_ laser vaporization was associated with higher AFC postoperatively compared to cystectomy [28]. Moreover, sutures or applying hemostatic sealants for hemostasis were more effective in preserving OR compared to bipolar coagulation [116]. Notably, the adverse effect of encapsulation of fluid and foreign body granulomatous reaction was reported in some cases [180].

Interestingly, a recent study investigated the optimal time for laparoscopic cystectomy in patients with unilateral OMA, which showed that the serum AMH levels decreased more remarkably in patients who received surgery at the late luteal phase compared with those at the early follicular phase [181]. The authors hypothesized that the border between OMA and normal ovarian tissue might loosen inflammatory edema of the tissue during the late luteal phase, making it easier to remove the cyst wall and reducing the loss of the normal ovarian cortex. More research is needed to explore optimal surgical options for endometriosis that would not lead to reduced OR.

#### 3.5.3. Therapeutic Targets to the Signaling Pathways Leading to Reduced OR in Women with Endometriosis

Therapeutic targets aimed at reversing ovarian function are pivotal for the development of novel drugs for patients with endometriosis. Based on the potential mechanisms leading to reduced OR in endometriosis described in part 5, 5 drugs targeting the potential signaling pathways were screened from the Therapeutic Target Database and listed in Table 3 [182]. These drugs are already available in the market for other diseases, while they could also provide a possibility for their application in restoring OR in women with endometriosis.

Besides these drugs, some novel therapies are under investigation and have the potential to restore OR. The overexpression of TIMP-1 was found to induce impaired folliculogenesis and oocyte quality in patients with endometriosis [19,20]. By inhibiting TIMP1 expression with function-blocking antibody, a rat model of endometriosis showed similar follicle, corpus luteum, zygote numbers, and embryo quality to sham rats. The findings indicated that neutralizing TIMP1 benefits further study on the way to developing a new endometriosis drug [19]. In another study using a rat model, lowered endometriosis scores were found following treatment with anastrozole and everolimus, the inhibitors of mTOR. Additionally, there is a significant reduction in ovarian follicles after anastrozole treatment but not everolimus treatment. These results support the hypothesis that everolimus has the potential to restore OR and improve fecundity in rats with endometriosis [17].

#### 3.5.4. The Natural Products to Treat Endometriosis That Will Not Lead to Ovulation Inhibition

As an estrogen-dependent inflammatory reproductive disease, endometriosis affects women of reproductive age. The present medical treatments for endometriosis mainly consist of hormonal therapy. However, hormonal therapy will lead to inhibition of ovulation, which is unsuitable for women with childbearing willingness. Thus, novel therapeutic targets that will not lead to ovulation inhibition are urgently needed.

Oleuropein is a natural product extracted from olive leaves [188], which can selectively inhibit Erβ activity in vitro [14]. The administration of Oleuropein effectively inhibited the growth of ectopic lesions in both mouse models and humans. This suppression was achieved by reducing proliferation and promoting apoptosis in endometriotic lesions. Additionally, Oleuropein was found to enhance fertility in mice with endometriosis [14]. Melatonin is a hormone synthesized by the pineal gland, which has been shown to reduce endometriosis lesions and relieve endometriosis-related pain in various studies [189,190]. It works by inhibiting cell proliferation, angiogenesis, oxidative stress, and so on [191]. Based on present evidence, melatonin supplements are believed to be safe for short-term use without reproductive toxicity [192]. Furthermore, melatonin can improve the oocyte quality in the IVF-ET system and ovarian function [193,194]. In mice receiving melatonin via drinking water at a concentration of 30 μg/mL for 21 days, there was a notable increase in the number of antral follicles compared to the control group. Additionally, the group treated with melatonin exhibited a significantly higher rate of hatched blastocyst formation compared to the control group [193]. In a prospective randomized controlled clinical trial, a total of 60 patients diagnosed with disturbed sleep were enrolled. These patients were divided into two groups: one group received 3 mg melatonin oral tablets starting from the 3rd to the 5th day of the menstrual cycle and continued until the day of human chorionic gonadotropin (hCG) injection for controlled ovarian hyperstimulation, while the other group did not receive melatonin. The group administered with melatonin exhibited significantly higher mean numbers of retrieved oocytes, mean counts of mature (MII) oocytes, and a greater ratio of G1 embryos when compared to the group that did not receive melatonin.

It indicates that melatonin may be a novel therapeutic intervention for endometriosis with less impact on OR. However, the long-term therapeutic efficacy of melatonin in treating endometriosis still needs to be elucidated.

## 4. Conclusions

Endometriosis, with its deeply rooted inflammatory characteristics, has emerged as a significant factor that negatively impacts OR and, consequently, the fertility potential of affected women. Using a comprehensive amalgamation of evidence gathered from in vitro, in vivo, and human studies, this review has shed light on the multifaceted consequences of this disorder. Notably, the decline in OR is not solely a result of the disease itself. Therapeutic interventions, which are aimed at managing or alleviating the condition, also contribute to the decrease. A critical point of disruption is the intrafollicular environment. The surge of cytokines and ROS, indicators of the disease’s inflammatory nature, set off a chain reaction that ultimately results in the fibrosis of the ovarian cortex. This altered environment adversely affects folliculogenesis, leading to impaired gamete development. Adding another layer to this complicity, recent research points to the hyperactivation of primordial follicles. This phenomenon, believed to be spurred by the inflammation and fibrosis linked to endometriosis, accelerates the progression of immature follicles, further reducing OR. Moreover, the review underscores the double-edged sword that therapeutic interventions present. In contrast, hormone treatments showed drawbacks such as hampers effective ovulation. Surgical interventions present their own set of challenges, such as the potential for vascular damage, inadvertent removal of healthy ovarian tissue, postoperative adhesions, and the risk of overstimulating primordial follicles.

However, it is important to acknowledge the complexity and multifactorial nature of endometriosis, such as diverse subtypes and stages among affected individuals, which makes it difficult to establish a clear causative relationship with reduced OR. The absence of standardized and precise measurements for OR further adds complexity to the interpretation of existing evidence. Moreover, our study has some limitations: (1) the non-homogeneous subtypes and stages of endometriosis in the studies included in this review; (2) heterogeneity between studies (i.e., study design, OR makers) included in the review; (3) limited number and quality of studies included in the review. These limitations underscore the need for further research to better elucidate the connections between endometriosis and OR, as well as the effects of different treatment modalities on this important reproductive parameter. Despite these challenges, our review highlights the significance of addressing OR in the context of endometriosis and offers potential strategies for refining treatment options to mitigate the risk of diminished OR in affected patients. The proposed modifications and emergent therapeutic strategies hold promise and open new avenues for the medical community to invest in further research to solidify these findings and refine the potential treatment methodologies.

## Figures and Tables

**Figure 1 ijms-24-15967-f001:**
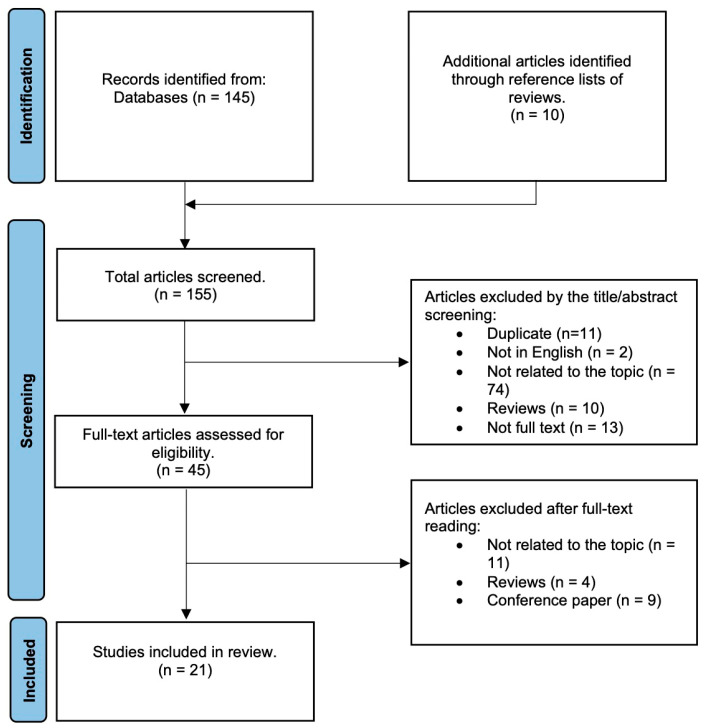
Flow diagram summarizing the study selection process.

**Figure 2 ijms-24-15967-f002:**
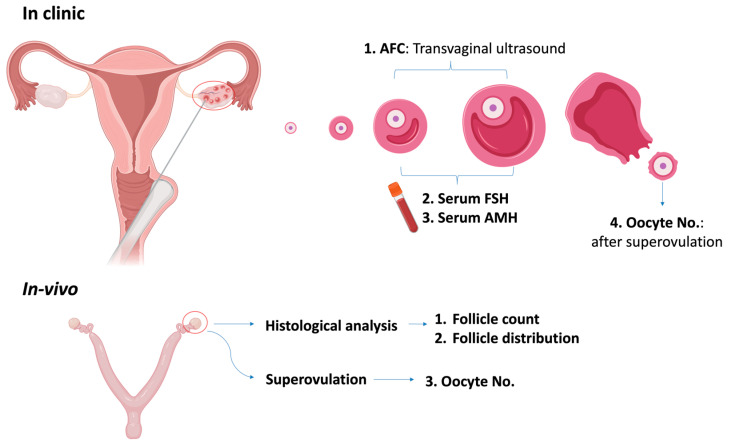
Quantitative assessment of ovarian reserve. AFC: antral follicle count; FSH: follicle-stimulating hormone; AMH: anti-Müllerian Hormone.

**Figure 3 ijms-24-15967-f003:**
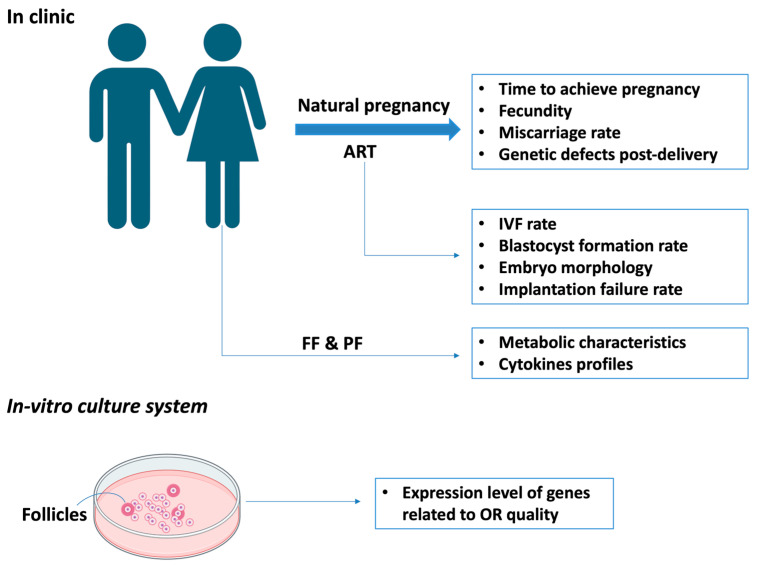
Qualitative assessment of ovarian reserve. ART: assisted reproductive technologies; FF: follicular fluid; PF: peritoneal fluid.

**Table 1 ijms-24-15967-t001:** Studies on the effect of endometriosis and its therapeutic interventions on ovarian reserve.

Author, Year	Study Type	Disease Subtype	Interventions	OR Maker
Kanellopoulos D et al., 2022 [13]	In vivo/rat	OMA/diffuse intraperitoneal endometriosis/subcutaneous endometriosis	-	The number of luteinized unruptured follicles ↑
Park Y et al., 2020 [14]	In vivo/mouse	Diffuse intraperitoneal endometriosis	Oleuropein (25 mg/kg/day, p.o.) for 21 days	Pregnancy rate (vehicle vs. oleuropein vs. no endometriosis = 70%:100%:100%)
Hayashi S et al., 2020 [15]	In vivo/mouse	OMA	-	The number of pups ↓
Takeuchi A et al., 2019 [16]	In vivo/mouse	Diffuse intraperitoneal endometriosis	-	Primordial follicle count ↓
Kacan T et al., 2017 [17]	In vivo/rat	Peritoneal endometriosis	Everolimus (1.5 mg/kg/day, p.o.) for 14 days	Follicle number ↑
Ozer H et al., 2013 [18]	In vivo/rat	Peritoneal endometriosis	Retinoic acid (10 mg/kg/day, p.o.) for 10 days	Primordial follicle count ↑
Stilley JA et al., 2010 [19]	In vivo/rat, in vitro	DIE	TIMP1-blocking antibody (32μg/kg, i.p., every third day for a total of three injections)	Follicle number ↑ Embryo quality ↑
Stilley JA et al., 2009 [20]	In vivo/rat, in vitro	DIE	-	Antral follicle number ↓ Oocyte number ↓ Oocyte quality ↓ Zygotes number ↓ Embryo quality ↓ Fecundity ↓
D’Hooghe TM et al., 1996 [21]	In vivo/baboon	Diffuse intraperitoneal endometriosis	-	Cycle pregnancy rate (stage II and stage III-IV) ↓
Barragán JC et al., 1992 [22]	In vivo/rat	Peritoneal endometriosis/DIE	-	Pregnancy rate ↓
Schenken RS et al., 1991 [23]	In vivo/monkey	Peritoneal endometriosis	-	Pregnancy rate (moderate or severe disease) ↓
Muraoka A et al., 2021 [24]	Human study	OMA	Laparoscopic ovarian cystectomy with dienogest (2 mg/day) or buserelin acetate (1.8 mg/month) for 2 months preoperatively and 2 months postoperatively	AMH after 1-year follow-up (dienogest group ↑ vs. buserelin group ↓)
Sireesha MU et al., 2021 [25]	Human study	OMA	Laparoscopic ovarian cystectomy	AMH (1 week and 3 months postoperatively) ↓
Barra F et al., 2020 [26]	Human study, in vitro	OMA	Dienogest (2 mg/day, p.o.) for 3 months	Implantation rate ↑ Pregnancy rate ↑ Live birth rate ↑ The number of oocytes retrieved, two-pronuclear embryos and blastocysts ↑
Karadağ C et al., 2020 [27]	Human study	OMA	-	AMH ↓ AFC ↓
Candiani M et al., 2018 [28]	Human study	OMA	CO_2_ laser vaporization or laparoscopic ovarian cystectomy	AFC (laser vaporization group ↑ vs. cystectomy group ↓) and AMH (laser vaporization group ↓ vs. cystectomy group↑) at 3-month follow-up
Kasapoglu I et al., 2018 [29]	Human study	OMA	-	AMH ↓
Muller V et al., 2017 [30]	Human study, in vitro	OMA	Laparoscopic ovarian cystectomy followed by dienogest (2 mg/day, p.o.) treatment for 6 months	Pregnancy rate ↑ Live birth rate ↑ The number of oocytes retrieved and A-class embryos ↑
Taniguchi F et al., 2015 [31]	Human study	OMA	OCP (DRSP/EE 3 mg/20μg, 24/4 regimen, 6 cycles)	AMH (after 6 cycles) ↓
Lima ML et al., 2015 [32]	Human study	OMA	-	AFC ↓ The number of oocytes retrieved (-)
Li CZ et al., 2009 [33]	Human study	OMA	Electrocoagulation or suture after excision of OMA	AFC and ovarian diameter at 3-, 6-, 12-month follow-up (electrocoagulation ↓ vs. suture ↑)

↑: increased, ↓: decreased. Abbreviations: OMA: endometrioma, DIE: deep infiltrating endometriosis, OCP: oral contraceptive pills, DRSP: drospirenone, EE: ethinylestradiol, AMH: anti-Mullerian hormone, AFC: antral follicle count.

**Table 2 ijms-24-15967-t002:** The comparison of the existing measurements of ovarian reserve in the clinic.

Measurement	Side of Ovary	Follicle Stage	Menstrual Cycle-Sensitive	Expense
AFC	One-by-one	Antral follicle	No	**
FSH	Both	Developing follicles	Yes	*
AMH	Both	Preantral and antral	No	*
Oocyte retrieval after superovulation	Both/Unilateral	/	/	***

*: <$100; **: $100–1000; ***: >$1000.

**Table 3 ijms-24-15967-t003:** Potential drugs targeting the signaling pathways and molecules of endometriosis-related to reduced ovarian reserve.

Signaling Pathway/Molecules	Drug ^1^	Drug Type	Indication	Clinical Phase ^2^	Pregnancy Category	Potential Long-Term Implications ^3^
IL-8	BMS-986253	Monoclonal antibody	Coronavirus Disease 2019	Phase 2	Unknown	Grade 1/2 [183]
IL-12	GEN-1	Gene therapy	Ovarian cancer	Phase 2	Unknown	Grade 1, 2, 3 [184]
TNF-α	ADVAX	Highly potent adjuvants	Human immunodeficiency virus infection	Phase 1	B	Grade 1 [185]
ROS	Tafenoquine	Small molecular drug	Malaria; Plasmodium vivax malaria	Approved	C	Grade 1, 2 [186]
PI3K/AKT/mTOR pathway (PAm pathway)	Sirolimus	Small molecular drug	Multiple myeloma; Organ transplant rejection	Approved	C	Grade 1, 2, 3 [187]

^1^ Data were extracted from the Therapeutic target database. ^2^ Data were extracted from clinicaltrials.gov (accessed on 30 July 2023) (NCT04347226, NCT05739981, NCT00902824). ^3^ Adverse event grade is according to the National Cancer Institute Common Terminology Criteria for Adverse Event v5.0.

## Data Availability

No new data were created in this study. All the data reported in this review were found in original articles cited in the text.

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
