# Peer review of "Diminished Ovarian Reserve in Endometriosis: Insights from In Vitro, In Vivo, and Human Studies—A Systematic Review"

_ijms, 2023, doi:10.3390/ijms242115967_

Round 1

Reviewer 1 Report

1. Title Clarity: The title is comprehensive but might benefit from a more concise phrasing. Consider: "Diminished Ovarian Reserve in Endometriosis: Insights from In-vitro, In-vivo, and Human Studies."

2. Abstract Structure: The abstract provides a clear overview of the study's objectives and findings. However, it might be beneficial to include a brief mention of the primary methodologies used to give readers an immediate understanding of the research approach.

3. Depth of Introduction: The introduction on ovarian reserve is informative. It might be helpful to provide more background on the significance of the ovarian reserve in the broader context of reproductive health.

4. Methodological Clarity: The section on predicting the ovarian reserve provides a detailed overview of the methods. However, it would be beneficial to include more information on the advantages and disadvantages of each method, especially in the context of endometriosis research.

5. Discussion on Endometriosis: The link between endometriosis and reduced ovarian reserve is well-presented. It would be valuable to delve deeper into the molecular or cellular mechanisms that might be at play, even if they are speculative.

6. Treatment Section: The section on endometriosis treatments and their potential impact on ovarian reserve is crucial. Consider expanding on the long-term implications of these treatments and any alternative or emerging treatments that might have a lesser impact on the ovarian reserve.

7. Figures and Tables: While the provided content mentions a "Graphical Abstract," it would be beneficial to have more figures, tables, or charts that visually represent the key findings, especially the quantitative data on ovarian reserve metrics.

8. Conclusion: Ensure that there is a robust conclusion section that summarizes the key findings, their implications, and potential future research directions.

9. References: Ensure that all statements, especially those related to statistics and previous research findings, are adequately cited. It would also be beneficial to include a comprehensive list of references at the end of the paper.

Abstract: The abstract is well-structured and provides a clear overview. However, consider rephrasing the sentence "This review summarizes evidence from in-vitro, in-vivo, and human studies about the diminished ovarian reserve in endometriosis" for better clarity. Suggestion: "This review consolidates evidence from in-vitro, in-vivo, and human studies on the diminished ovarian reserve associated with endometriosis."

Section 2 - The prediction of ovarian reserve: The sentence "The quantitative markers reflect the number of follicles and oocytes, while the latter represents the quality of oocytes during folliculogenesis and after ovulation" might be clearer if rephrased. Suggestion: "Quantitative markers indicate the number of follicles and oocytes, whereas qualitative markers represent oocyte quality during folliculogenesis and post-ovulation."

Section on Research: The sentence "Research wise, sixteen genes encoding including GRM3, UTS2, FAM9B, and LYG2, which encode for secretory proteins, have been pinpointed due to their unique expression" could be restructured for clarity. Suggestion: "From a research perspective, sixteen genes, including GRM3, UTS2, FAM9B, and LYG2, which encode secretory proteins, have been identified for their unique expression."

Author Response

Response to Reviewer 2 Comments

1. Summary

Dear Reviewer,

I would like to express my sincere gratitude for the time and effort you have dedicated to providing the valuable comments and suggestions. We have incorporated most of the suggestions and provided a point-by-point response, which is listed below. As we have made significant revision, we also tracked all the changes made within the manuscript.

Comments 1. Title Clarity: The title is comprehensive but might benefit from a more concise phrasing. Consider: "Diminished Ovarian Reserve in Endometriosis: Insights from In-vitro, In-vivo, and Human Studies."

Response 1: Thank you for the valuable suggestions. We changed the title to "Diminished Ovarian Reserve in Endometriosis: Insights from In-vitro, In-vivo, and Human Studies.".

Comments 2. Abstract Structure: The abstract provides a clear overview of the study's objectives and findings. However, it might be beneficial to include a brief mention of the primary methodologies used to give readers an immediate understanding of the research approach.

Response 2: We appreciate your insightful comments and have made the necessary revisions accordingly. We added a brief mention of the primary methodologies in the abstract “we investigated articles with the related keywords, which were published in recent years.” And we added the search strategy in P2L42-49 of the manuscript.

Comments 3. Depth of Introduction: The introduction on ovarian reserve is informative. It might be helpful to provide more background on the significance of the ovarian reserve in the broader context of reproductive health.

Response 3: Thank you for the constructive feedback. We added more background on the significance of the ovarian reserve in the broader context of reproductive health in P3L64-66: ‘Given OR represents the reproductive potential of women, OR test can provide additional information when tailoring the treatment plan for infertile couples, as well as guide fertility preservation.’

Comments 4. Methodological Clarity: The section on predicting the ovarian reserve provides a detailed overview of the methods. However, it would be beneficial to include more information on the advantages and disadvantages of each method, especially in the context of endometriosis research.

Response 4: Thank you for the constructive feedback. In section 4. the prediction of ovarian reserve, we added more information on the advantages and disadvantages of each method, and listed the comparison of the existed measurements of ovarian reserve in clinic in table 1.

Comments 5. Discussion on Endometriosis: The link between endometriosis and reduced ovarian reserve is well-presented. It would be valuable to delve deeper into the molecular or cellular mechanisms that might be at play, even if they are speculative.

Response 5: Thank you for the valuable suggestions. In Section 5, we mainly discussed the link between endometriosis and reduced ovarian reserve based on different studies. And the molecular or cellular mechanisms that might play roles in the decreased ovarian reserve in patients with endometriosis are discussed deeply in section 6, for example inflammation, oxidative stress, fibrosis, and DNA damage, dysregulated ovulation and ovarian steroids secretion, and hyperactivation of primordial follicle. Furthermore, we separated the studies in each potential mechanism into human studies, in-vitro, in-vivo studies for a better understanding in each subsection.

Comments 6. Treatment Section: The section on endometriosis treatments and their potential impact on ovarian reserve is crucial. Consider expanding on the long-term implications of these treatments and any alternative or emerging treatments that might have a lesser impact on the ovarian reserve.

Response 6: Thank you for the valuable suggestions. We have taken the your comments into account and made the appropriate modifications to address your concerns. For the potential therapeutic interventions, we have added the long-term implications according to National Cancer Institute Common Terminology Criteria for Adverse Event V5.0 in table 2. And the alternative or emerging treatments, for instance the modification of laparoscopic cystectomy for endometriosis, and natural products that have a lesser impact on OR, were described in section 7.

Comments 7. Figures and Tables: While the provided content mentions a "Graphical Abstract," it would be beneficial to have more figures, tables, or charts that visually represent the key findings, especially the quantitative data on ovarian reserve metrics.

Response 7: We are grateful for your valuable input, which allowed us to refine our manuscript. In order to have a better representation, we added Fig 1 and Fig 2 for the quantitative and qualitative assessment of ovarian reserve. Moreover, we summarised the existed qualitative assessments in clinic in table 1.

Comments 8. Conclusion: Ensure that there is a robust conclusion section that summarizes the key findings, their implications, and potential future research directions.

Response 8: Thank you for your feedback. The conclusion section is thoroughly revised and the currently provided content summarizes the key findings, discusses their implications, and suggests potential future research directions related to endometriosis and ovarian reserve accordingly.

Comments 9. References: Ensure that all statements, especially those related to statistics and previous research findings, are adequately cited. It would also be beneficial to include a comprehensive list of references at the end of the paper.

Response 9: Thank you. We have thoroughly reviewed your feedback and have revised our manuscript accordingly. We double checked the references and make sure all the statements are adequately cited. The full references are listed at the end of the paper.

Comments on the Quality of English Language:

Abstract: The abstract is well-structured and provides a clear overview. However, consider rephrasing the sentence "This review summarizes evidence from in-vitro, in-vivo, and human studies about the diminished ovarian reserve in endometriosis" for better clarity. Suggestion: "This review consolidates evidence from in-vitro, in-vivo, and human studies on the diminished ovarian reserve associated with endometriosis."

Response: Thank you for the valuable suggestions. We have revised the sentence and replaced by "This review consolidates evidence from in-vitro, in-vivo, and human studies on the diminished ovarian reserve associated with endometriosis."

Section 2 - The prediction of ovarian reserve: The sentence "The quantitative markers reflect the number of follicles and oocytes, while the latter represents the quality of oocytes during folliculogenesis and after ovulation" might be clearer if rephrased. Suggestion: "Quantitative markers indicate the number of follicles and oocytes, whereas qualitative markers represent oocyte quality during folliculogenesis and post-ovulation."

Response: We acknowledge your helpful comment. We have revised the sentence and changed the interpretation accordingly to make it clearer.

Section on Research: The sentence "Research wise, sixteen genes encoding including GRM3, UTS2, FAM9B, and LYG2, which encode for secretory proteins, have been pinpointed due to their unique expression" could be restructured for clarity. Suggestion: "From a research perspective, sixteen genes, including GRM3, UTS2, FAM9B, and LYG2, which encode secretory proteins, have been identified for their unique expression."

Response: Thank you for the valuable suggestions. We have restructured the sentence to make it clearer.

Reviewer 2 Report

This paper aimed to enhance our understanding of whether and how endometriosis, as well as its interventions, contribute to the reduction of ovarian reserve, as authors state..

This is a narrative review, but it lacks a structured format and reporting. I would suggest the reformation into a proper review with all sections to be referred accordingly, such as aim/rationale, search strategy, and so on, including limitations.

Minor suggestions include:

1.      Revisiting the definition of ovarian reserve.

2.      The structure of the different categories of the sub sections, should be revisited also, in a logical sense.

3.      The length of the sub sections should be revisited, and distributed equally to each objective.

4.      All  in-vitro, in-vivo, and human studies should be separated in reporting.

5.      The number of references is very big. I would suggest this to be shortened, including only the relevant systematic reviews and those papers that contribute substantially towards the supporting of each section.

Mild changes are needed.

Author Response

1. Summary

Dear Reviewer,

I would like to express my sincere gratitude for the time and effort you have dedicated to providing the valuable comments and suggestions. We have incorporated most of the suggestions and provided a point-by-point response, which is listed below. As we have made significant revision, we also tracked all the changes made within the manuscript.

This paper aimed to enhance our understanding of whether and how endometriosis, as well as its interventions, contribute to the reduction of ovarian reserve, as authors state.

This is a narrative review, but it lacks a structured format and reporting. I would suggest the reformation into a proper review with all sections to be referred accordingly, such as aim/rationale, search strategy, and so on, including limitations.

Response: Thank you for the valuable suggestions. We have thoroughly reviewed your feedback and have revised our manuscript accordingly. The manuscript have been reformed more properly, and the aim, search strategy were added as part 1, 2. We added the limitation in the conclusion part, P16L676-L680.

Minor suggestions include:

Comments 1.      Revisiting the definition of ovarian reserve.

Response 1: Thank you for the valuable suggestions. We revised the definition of ovarian reserve in the section 3. P2L52-56: ‘Ovarian reserve (OR) has long been defined as the number and quality of oocytes a woman possesses, which determine her reproductive potential. However, the most updated conception of OR only refers to the total number of oocytes present in the ovary, i.e., oocyte quantity, which is different from oocyte quality relating to a fertilized oocyte's potential for a live birth.’.

Comments 2.      The structure of the different categories of the sub sections, should be revisited also, in a logical sense. 

Response 2: We are grateful for your valuable input. We revised the structure of different sections and restructured them in a more logical sense.

Comments 3.      The length of the sub sections should be revisited and distributed equally to each objective.

Response 3: We have taken the your comments into account and made the appropriate modifications to address your concerns. While in specific sections, such as 5.1.1 and 5.1.2, there is an unequal distribution due to the multitude of endometriosis treatment types, each requiring elaboration. And in 4.1 and 4.2, the number of literature encompassing quantitative assessment and qualitative assessment of ovarian reserve is diverse, creating a hurdle in attaining consistent content length. Nonetheless, we have tried to make the descriptions shorter to reduce verbosity.

Comments 4.      All in-vitro, in-vivo, and human studies should be separated in reporting.

Response 4: We acknowledge your helpful comment. We have revised the full manuscript and report in-vitro, in-vivo, and human studies separately especially in section 4, section 5, and section 6.

Comments 5.      The number of references is very big. I would suggest this to be shortened, including only the relevant systematic reviews and those papers that contribute substantially towards the supporting of each section.

Response 5: Thank you for your feedback. We thoroughly checked the references and make sure all the statements are adequately cited and deleted the irrelevant references. So far we shorted the references within 190.

Comments on the Quality of English Language: Mild changes are needed.

Response: Thank you for your feedback. We’ve checked the language and revised it to make the interpretation more properly.

Round 2

Reviewer 1 Report

Accept in present form

Author Response

Dear Reviewer,

I would like to express my sincere gratitude for the time and effort you have dedicated to providing the valuable comments and suggestions to our manuscript. We have diligently incorporated the majority of the suggested changes into our manuscript for the first round, striving to enhance its overall quality and clarity.

We believe that these revisions according to your prior comments have significantly improved the clarity and structure of our manuscript, aligning it more closely with the standards of the International Journal of Molecular Sciences. We are grateful for your guidance throughout this process and look forward to your feedback on our revised submission.

Thank you once again for your consideration.

Yours sincerely,

TAN, Zhouyurong, B.Sc (Hons), M.B.B.S.

PhD student

Department of Obstetrics & Gynaecology

The Chinese University of Hong Kong

Reviewer 2 Report

I think that authors made a substantial effort to improve their manuscript. There are still points to be revisited, at least in terms of structure.

I would recommend authors to:

1. take examples from how a systematic review is written e.g. and follow the structured steps (not the substance).

2. write the introduction section in the beginning

3. under the aim, please refer to what will be the points to be addressed

4. when I meant limitations, I also meant limitations because of the quality of the study itself. 

adequate 

Author Response

Dear Reviewer,

I would like to express my sincere gratitude for the time and effort you have dedicated to providing the valuable comments and suggestions. We have diligently incorporated the majority of the suggested changes into our manuscript, striving to enhance its overall quality and clarity. As part of our commitment to transparency and thoroughness, we have meticulously tracked all changes made within the manuscript.

Below, we provide a point-to-point response addressing the changes we have made in accordance with you and the editors’ feedback:

Comments 1: take examples from how a systematic review is written e.g. and follow the structured steps (not the substance).

Response 1: We sincerely appreciate your insightful comments and have restructured our manuscript accordingly. We referred to examples of systematic reviews published in IJMS and other journals, and now our manuscript follows a structured format. It begins with an introduction, followed by the methodology and results, and concludes with a comprehensive summary. We also provided the Flow diagram summarizing the study selection process according to PRISMA guidelines, while could be found in Fig.1.

Comments 2: write the introduction section in the beginning

Response 2: Thank you for this valuable suggestion. We have added the introduction as the opening section of our manuscript. This introduction provides an overview of endometriosis, its impact on fertility and ovarian reserve, the existing controversies in current literature, and the specific goals of our review. The introduction can be found in P2L35-P3L82.

Comments 3. under the aim, please refer to what will be the points to be addressed

Response 3: Your constructive feedback has been duly noted. In response, we have included a detailed section under the aim that outlines the specific points addressed in our manuscript. This addition is designed to engage the reader and provide a clear understanding of our objectives. It can be found in P2L78-P3L82:

“In detail, to (1) describe the existing prediction methods of ovarian reserve and evaluate the advantages and disadvantages of each method; (2) elucidate the relevance of endometriosis per se and its treatments and reduced ovarian reserve, and reveal the potential mechanisms; (3) highlight the potential therapeutic interventions for preserve ovarian reserve in patients with endometriosis.”

Comments 4. when I meant limitations, I also meant limitations because of the quality of the study itself. 

Response 4: Thank you for this valuable suggestion. We have critically evaluated the limitations of the included studies and our review.

It can be found in P18L735-P18L738:

“(1) the non-homogeneous subtypes and stages of endometriosis in the studies included in this review; (2) heterogeneity between studies (i.e., study design, OR makers) included in the review; (3) limited number and quality of studies included in the review.”

We believe that these revisions have significantly improved the clarity and structure of our manuscript, aligning it more closely with the standards of the International Journal of Molecular Sciences. We are grateful for your guidance throughout this process and look forward to your feedback on our revised submission.

Thank you once again for your consideration.

Yours sincerely,

TAN, Zhouyurong, B.Sc (Hons), M.B.B.S.

PhD student

Department of Obstetrics & Gynaecology

The Chinese University of Hong Kong
